# Assessment of Myocardial Diastolic Dysfunction as a Result of Myocardial Infarction and Extracellular Matrix Regulation Disorders in the Context of Mesenchymal Stem Cell Therapy

**DOI:** 10.3390/jcm11185430

**Published:** 2022-09-15

**Authors:** Patrycja Piątek-Matuszak, Robert Pasławski, Urszula Pasławska, Liliana Kiczak, Michał Płóciennik, Adrian Janiszewski, Marcin Michałek, Adrian Gwizdała, Jarosław Kaźmierczak, Jarosław Gorący

**Affiliations:** 1Department of Cardiology, Pomeranian Medical University, 70-204 Szczecin, Poland; 2Doctoral Study, Pomeranian Medical University, 70-204 Szczecin, Poland; 3Veterinary Institute, Faculty of Biological and Veterinary Sciences, Nicolaus Copernicus University, 87-100 Toruń, Poland; 4Department of Biochemistry and Molecular Biology, Faculty of Veterinary Medicine, Wroclaw University of Environmental and Life Sciences, 50-375 Wrocław, Poland; 5Department of Internal Disease and Veterinary Diagnosis, Faculty of Veterinary Medicine and Animal Sciences, Poznań University of Life and Sciences, 60-637 Poznań, Poland; 6Department of Internal Medicine with Clinic of Diseases of Horses, Dogs, and Cats, Faculty of Veterinary Medicine, Wrocław University of Environmental and Life Sciences, 50-375 Wrocław, Poland; 7First Department of Cardiology, Poznan University of Medical Sciences, 61-701 Poznań, Poland; 8Independent Laboratory of Invasive Cardiology, Pomeranian Medical University in Szczecin, 70-204 Szczecin, Poland

**Keywords:** diastolic dysfunction, myocardial infarction, extracellular matrix, stem cell

## Abstract

The decline in cardiac contractility due to damage or loss of cardiomyocytes is intensified by changes in the extracellular matrix leading to heart remodeling. An excessive matrix response in the ischemic cardiomyopathy may contribute to the elevated fibrotic compartment and diastolic dysfunction. Fibroproliferation is a defense response aimed at quickly closing the damaged area and maintaining tissue integrity. Balance in this process is of paramount importance, as the reduced post-infarction response causes scar thinning and more pronounced left ventricular remodeling, while excessive fibrosis leads to impairment of heart function. Under normal conditions, migration of progenitor cells to the lesion site occurs. These cells have the potential to differentiate into myocytes in vitro, but the changed micro-environment in the heart after infarction does not allow such differentiation. Stem cell transplantation affects the extracellular matrix remodeling and thus may facilitate the improvement of left ventricular function. Studies show that mesenchymal stem cell therapy after infarct reduces fibrosis. However, the authors did not specify whether they meant the reduction of scarring as a result of regeneration or changes in the matrix. Research is also necessary to rule out long-term negative effects of post-acute infarct stem cell therapy.

## 1. Introduction

Ischemic heart disease is one of the most common causes of death worldwide [1]. Its most dramatic and dangerous manifestation is myocardial infarction, commonly known asa heart attack. In 2018, cardiological organizations (European Society of Cardiology [ESC], American College of Cardiology Foundation [ACCF], American Heart Association [AHA], World Heart Federation [WHF]), based on objective markers of myocardial necrosis, agreed on the 4th universal definition of myocardial infarction. The essence of the diagnosis was the occurrence of acute myocardial damage and demonstration of its ischemic basis [2]. Left ventricular systolic dysfunction is the most common, but not the only, cause of post-infarction heart failure. Primary decline in contractility due to damage or loss of cardiomyocytes is supported and enhanced by changes in the heart’s extracellular matrix (ECM), leading to heart remodeling [3,4]. The term ‘matrix’ itself refers to the tissue component and more precisely to the substance produced by the cells and filling the space between them. An excessive ECM response in the case of ischemic cardiomyopathy may contribute to an elevated fibrotic compartment and finally diastolic dysfunction [5,6].

The objective of the literature review was to determine whether treatment with mesenchymal stem cells after myocardial infarction has a negative, neutral or beneficial effect on diastolic function and regulation of the extracellular matrix.

### 1.1. Mechanisms Underlying Diastolic Dysfunction after MI

Increased fibrosis was observed in both the infarct zone (IZ) and the non-infarct zone (NIZ). In the infarct zone, successive reparative fibrosis (also called corrective, replacement fibrosis) was observed [5,7]. In the process, connective tissue fills the gaps after dead myocytes and is limited to one area of the tissue. In the non-infarct zone, there has been observed reactive, disseminated hypertrophy, connected with interstitial fibrosis. This process causes excessive and diffuse accumulation of connective tissue in interstitial and perivascular spaces [3,8]. During the reactive fibrosis, no loss of myocytes is observed, and the scope of the remodeling process comprises the perivascular space and interstitial tissue. This type of fibrosis occurs more frequently in patients with heart failure caused by hypertension, hypertrophic cardiomyopathy or aortic stenosis than after a myocardial infarction [9].

Fibroproliferation is a tissue defense aimed at quickly closing the damaged area and maintaining the integrity of the heart. Balance in this process is of paramount importance. The reduced post-infarction response causes scar thinning and more pronounced left ventricular remodeling, while excessive fibrosis (as mentioned above) leads to impairment of heart function. The process of excessive fibrosis results from increased collagen synthesis, which outweighs unchanged or reduced degradation [3,10,11]. Fibroblasts and myofibroblasts separate extracellular procollagen chains that connect with each other and cross-link to form collagen fibrils [5,12]. Two types of collagen dominate the heart: 80% type I collagen, which makes the heart muscle resistant to deformation due to the thickness of the fibers, and type III collagen (11%), whose thin fibers give the heart elasticity. The remaining forms of collagen account for 9% of its total amount [5,13,14]. Collagen cross-linking increases myocardial tensile strength and collagen fibers’ resistance to degradation by matrix metalloproteinases [3,15,16]. The pathological change in the collagen matrix is caused by a disorder of homeostasis of pro- and anti-fibrotic factors (cytokines, chemokines, growth factors, proteases, hormones, reactive forms of oxygen) [17]. Imbalance in collagen metabolism occurs in myofibroblasts, which are formed from fibroblasts as a result of the expression of alpha-smooth muscle actin (characteristic of smooth muscle cells), as well as the appearance of an extensive, synthetically active reticuloendoplasmic area [3,4,18]. Myocardial infarction causes successive changes in connective tissue, the pathophysiologically of which can be divided into four stages. Pathophysiological stages of the healing process after a myocardial infarction are presented in Figure 1.

Cardiac insult rapidly activates the matrix metalloproteinases (MMPs), leading to ECM fragmentation. These fragments act as bioactive pro-inflammatory matrikines [19]. In the very early, acute phase of the evolution of myocardial infarction (which covers 1 day), an increase in ECM-degrading proteases–matrix metalloproteinases (MMPs)–was observed. The increased activity of metalloproteinases, including MMP-2 and MMP-9, is preceded by an increase in transcription of MMP genes into pro-MMP, which is stimulated by IL-1, platelet-derived growth factor (PDGF) and TNF-α. This process is inhibited by TGF-β, retinoids, heparin and corticosteroids [5,20]. It is worth emphasizing the role of CD68+ macrophages, which, under ischemic conditions, enhance fibrosis and angiogenesis, and produce MMPs [21]. Macrophages also remove dead cells and ECM debris (in the first, inflammatory phase of infarction healing) and induce the release of anti-inflammatory mediators and proliferative factors, thus allowing the transition to the proliferative phase [22]. After 2 weeks, there are increased levels of tissue (endogenous) MMP inhibitors (TIMP), profibrotic aldosterone and angiotensin II, collagen, fibroblasts and growth factors (TGF beta and CTGF). The time frames observed for these events are well established in the mouse model, but studies in dog models show differences, such as a prolonged cell infiltration period and slower granulation formation after MI [23]. Late healing follows a period of relative collagen stabilization and leads to scar formation after about 3–6 weeks. This final, maturation phase varies in response to loading stimuli, including metabolic dysfunction, pressure load, genetic factors and aging [24].In this phase, the fibroblasts become quiescent, and the ECM collagen is cross-linked [25].The anti-inflammatory suppression of inflammation in the infarction zone does not apply to non-infarct segments where increased wall loading can locally activate cardiomyocytes, macrophages and fibroblasts, elevating MMPs and triggering chronic remodeling of the ECM [26].Earlier studies showing the association of MMPs (MMP2, MMP9, MMP14) with the development of post-infarction cardiomyopathy suggested that MMPs are responsible for this phenomenon [27].However, subsequent observations indicated that myocardial segments displaying decreased ECM activity did not regain function after revascularization [28].Thus, it seems that activation of the interstitial environment, enrichment of the number of growth factors, cells and matrix proteins may be necessary for regeneration of ischemic areas of the myocardium; however, actions dependent on ECM and fibroblasts may not be sufficient to activate the regeneration program [24].

Fibroproliferation in the infarct zone ispresented in Figure 2.

In the last stage, 1.5 months and later, fibrosis also develops in the NIZ zone [5,29]. In the following weeks, ECM degradation in the non-infarct zone (NIZ), loss of fibril collagen, and loss of myocytes due to apoptosis contribute to progressive global expansion and dysfunction of the left ventricle [5,29,30]. Notably, the infarct zone may widen as border zone (BZ) cells also contain cells at risk of death. This was proven by the increased level of T17-PLB phosphorylation and caspase-3 activation [31]. Long-term stimulation with pro-inflammatory cytokines results in the formation of diffuse small inflammation, scars and the transition to further remodeling processes. Collectively, data show that the ECM plays an important role in these processes [5,6,32]. Evidence suggests that some patients with preserved systolic function and diastolic dysfunction have normal collagen volume fractions despite myocardial stiffness, comparable to those with increased collagen volume fractions. In these patients, fibrosis coexists with comorbidities (e.g., diabetes) that affect the disease process [33]. Clinical and physiological features of diastolic dysfunction are altered after MI. Different pathophysiological responses in the IZ, BZ and NIZ result in a different clinical picture. Moreover, this picture has changed in recent years, as more and more people survived the myocardial infarction. They develop post-infarction heart failure in consequence of modern intervention procedures [34]. The gold standard in the diagnosis and assessment of cardiac fibrosis is the histopathological evaluation of myocardial biopsy samples. However, less invasive methods are constantly being sought. A number of circulating biomarkers have been proposed and analyzed, to observe myocardial injury, natriuretic peptides (ANP, BNP, NT-proBNP), troponin, and to analyze fibrinolytic activity: metalloproteinase-1 (MMP-1) and growth factors, e.g., transforming growth factor beta 1 (TGF-β1), promoting transactivation of myofibroblasts and ECM synthesis [3,35].

### 1.2. Diagnosis of Diastolic Dysfunction

Ultrasonography is a very helpful diagnostic tool. It enables the monitoring of morphological and functional changes of the heart muscle, along with the assessment of the regional systolic and diastolic functions of the left ventricle. Assessment of left ventricular diastolic dysfunction (LVDD), according to the guidelines of the American Society of Echocardiography (ASE) and the European Association of Cardiovascular Imaging (EACVI), is mainly based on six indicators: E-wave, E/A ratio, septal or lateral e′, mean E/e′, left atrial index volume (LAVI) and peak tricuspid velocity (TRpV) [36,37]. These indicators are presented in Figure 3.

However, other means, such as cardiac MRI and cardiac catheterization, can be more accurate in terms of quantitative assessment and accurate diagnosis of diastolic dysfunction and assessment of cardiac fibrosis. The diastolic function is closely related to the heart rate and rhythm, atrial function (mainly systolic), ventricular compliance, preload and atrioventricular valve function. Diastolic dysfunction is therefore a reflection of impaired left ventricular relaxation, resulting from its increased stiffness (advanced stages) and increased filling pressure (even more advanced stages) [36,38].

### 1.3. Stem Cell Therapy

The development of regenerative medicine and stem cell treatment of coronary heart disease has become a matter of interest for many scientists. It has been observed that, after myocardial infarction, the number of circulating endothelial progenitors CD34+ and CD133+ KDR and mesenchymal cells increases [39]. It has been demonstrated that, under normal conditions, the migration of progenitor cells to the lesion site and differentiation into heart cells occur, provided that the injury is minor [17,40,41]. In 2015, it was emphasized that in optimal conditions adult cardiac and marrow cells expressing protein C-kit (c-kit+) are able to generate new myocytes. These cells have the potential to differentiate into heart myocytes in vitro, but the changed micro-environment in the heart, being a result of infarction, doesnot allow such differentiation [42]. Mesenchymal stem cells (MSCs) enable physical contact between living cardiomyocytes, and they are the stimulants necessary to support the repair of the heart muscle. Moreover, as shown in the Central Illustration (Figure 4), mesenchymal stem cells seem to be a promising therapeutic agent for cardiac regeneration in myocardial diastolic dysfunction.

### 1.4. Impact of Stem Cell Therapy on Diastolic Function of the Heart

The key factor affecting the diastolic dysfunction of the myocardium as a result of myocardial infarction and dysregulation of the ECM in the context of MSC therapy is the type of stem cells supplied. Some stem cells seem to have greater differentiation capacity, while other may have greater paracrine activity or greater potential to stimulate neovascularization. It was reported that MSCs obtained from the subcutaneous white fat tissue of the abdomen of a pig (adipose tissue-derived mesenchymal stem cells [ATSC]), injected into a pre-planned area, improved left ventricular systolic function and reduced the scope of the infarction [43]. It also modulated the reconstruction of the local intercellular matrix, despite the fact that myocytes in BZ began to break down into necrotic tissues. Microscopic study helped to reveal that after ATSC injection myocytes as well as the vascular system remained undisturbed. Immunofluorescence staining proved spaces sparing the basal membrane with intact, smooth muscle layers and endothelium of blood vessels, ensuring blood flow. Another benefit of therapy (at the structural as well as metabolic level) is the observed increase in the number of formed vessels and better-preserved cardiomyocyte as well as Purkinje fibers [44]. Of note, most of the experiments that have strongly supported the beneficial effects of stem cell therapy have been carried out on young animals (rodents or pigs). It was a reasonable concern whether the response of heart tissue in elderly persons would be different due to the progressive decline of the heart’s ability to react with age [45]. However, clinical trials have dispelled these doubts by confirming that stem cell therapy profoundly increases functional recovery after myocardial infarction [46].

Many studies indicate a low immunogenicity of MSCs, thanks to which a single transplant is well tolerated by the recipient organism. However, repeated administration of MSCs may result in the production of all antibodies. Von Bonin et al., reported that the transplant of MSCs, which had contact with fetal bovine serum (FBS), induced the production of antibodies against FBS in the blood of the recipient. It is worth emphasizing that FBS is often used in the MSC culture medium. Thus, MSCs are dualistic in nature.

### 1.5. Influence of Stem Cell Therapy on Cardiac Fibrosis

Research shows that, thanks to the use of the latest pharmacological treatment methods on people with MI, we are able to eliminate clinical symptoms, but we are not able to eliminate fibrosis [3,47]. The resulting scar protects the heart muscle against rupture. On the other hand, chronic diffuse or focal reactive myocardial fibrosis resulting from pressure or volume overload exacerbates functional disorders [3,18]. Stem cell transplantation affects the remodeling of the extracellular matrix significantly, and this may contribute to the improvement of LV function. Most studies show that stem cell therapy after MI, combined with the administration of mesenchymal stem cells, reduces fibrosis. However, the authors of these studies did not specify whether they meant the reduction of post-infarction scar as a result of regeneration or changes in the extracellular matrix [46]. MSCs appear to significantly reduce the expression of type I and III collagen and significantly suppress the activity of the type III collagen promoter [47]. They reduce the expression of MMP-2 and MMP-9 proteins and increase the level of matrix modulating factors, such as metalloproteinase 2 (MMP-2), tissue inhibitors of matrix metalloproteinase (TIMP)-1 and TIMP-2, as well as matrix proteins thrombospondin-1 and tenascin C [48]. Paracrine substances derived from MSCs play a key role in reducing fibrosis, promoting neovascularization, modulating the extracellular matrix, cytoprotection, and inhibiting apoptosis and inflammation [49,50,51]. Some of the most important paracrine factors released by adult stem cells are growth factors. These are vascular endothelial growth factor VEGF, stromal cell origin factor 1 (SDF-1, also known as the CXC chemokine 1 motif CXCL12) and insulin-like growth factor-1 (IGF-1) [49,52,53,54,55,56].

Despite significant progress in this method of treatment, cell transplants still have an insufficient degree of implantation in the damaged heart and insufficient survival in the area of damage. In the years 2010–2020 several randomized and multicenter studies of the treatment of ischemic heart disease were carried out, using mainly autologous bone marrow-derived mononuclear cells, but also selected bone marrow cells. All these studies showed no additional benefit over standard therapy [57,58,59,60,61,62,63,64,65]. We still do not fully understand why clinical outcomes improve with combined stem cell MSC and CSC therapy without improving LV function or reducing scar size [66]. Some authors have suggested that the functional benefits of heart cell therapy result from an acute inflammatory wound healing response that rejuvenates the post-infarction area of the heart. This conclusion was based on the observation that there is an interesting effect of improving the work of the heart after administering cells killed by freezing and thawing. These cells induced regional macrophages CCR2+ and CX3CR1+ and provided functional rejuvenation of the heart following ischemia–reperfusion injury. This selective macrophage response altered the activity of cardiac fibroblasts, reduced the extracellular matrix content in the border zone and improved the mechanical properties of the damaged area [67].

### 1.6. Application of Fibroblasts in Stem Cell Therapy

An adult heart muscle has more fibroblasts compared to resident heart progenitor cells (cardiac fibroblasts make up more than 50% of all cells in the heart). Therefore, the concept was developed to increase the endogenous regenerative potential of an adult heart, performed by reprogramming fibroblasts with the use of retroviruses into ESC-like cells (referred to as induced pluripotential stem cells [iPS]) and then into new cardiomyocytes [61]. Using gene cocktails such as c-MYC, OCT3/4, SOX2 KLF4 or OCT3/4, SOX2, NANOG, LIN28, it was possible to reprogram human fibroblasts back intopluripotent stem cells [68,69]. Subsequent studies have shown similar results using fewer genes [70,71] and developed a non-viral methodology to avoid the potentially mutagenic effects of integrating viral delivery methods [72]. Today, generation of iPS cell lines is faster and cheaper, and promises to replace conventional sources of pluripotential stem cells.

### 1.7. Post-Infarction Treatment of Diastolic Dysfunction

Excessive cardiac fibrosis, associated with diastolic dysfunction, is still a clinical problem. There has been no uniform approach to remodeling therapy for several decades. Preventing post-infarction ECM remodeling seems to be very important, because it prevents such dangerous consequences as left ventricular dilatation and rupture [5,6,15,73]. The standard pharmacotherapy includes inhibitors of the angiotensin I convertase enzymes (ACE-I), aldosterone antagonists, blockers of the angiotensin II receptors and the β-adrenergic receptor blockers [5,32,74,75]. Patients are heterogeneous in pathophysiology and therapeutic response. The severity and characteristics of post-MI remodeling depend not only on the size of the acute MI, but also on age, sex, genetic background and concomitant disease. Some patients with hypertension or diabetes may show prolonged pro-inflammatory signaling after infarction, leading to dilative remodeling and systolic dysfunction. These patients may benefit from anti-inflammatory therapy [76]. In other diabetic patients, the hypertrophic and fibrotic reactions are more pronounced, leading to diastolic dysfunction. This type of unfavorable remodeling is associated with overactivity of the TGF-β system [77]. The identification of such patient subpopulations, using appropriate biomarkers or molecular imaging techniques, allows more effective therapies to be designed for MI patients [78].

### 1.8. Conclusions

In conclusion, cellular therapy for heart regeneration has much room for improvement before being considered an affordable, widely available, safe and effective clinical approach. It seems that studies should be carried out at this stage on animal models, as the effects of both non-hematopoietic and hematopoietic stem cells may differ in vitro and in vivo. Long-term research performed to rule out long-term negative effects of post-acute MI stem cell therapy, including diastolic failure, will also be necessary.

## Figures and Tables

**Figure 1 jcm-11-05430-f001:**
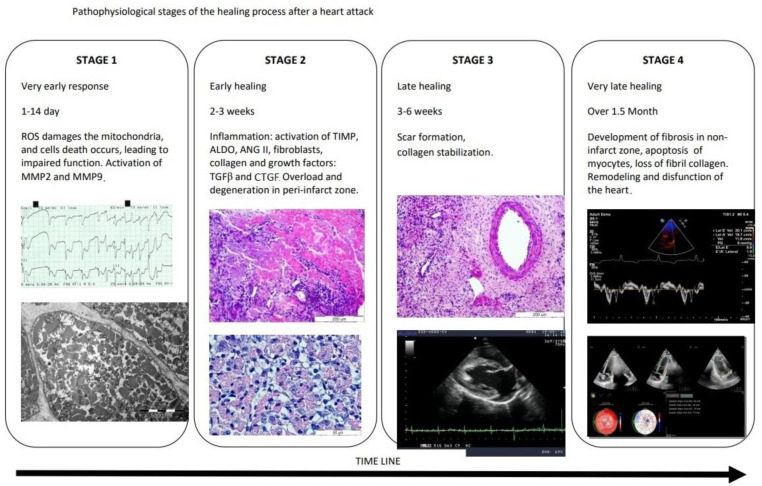
Pathophysiological stages of the healing process after a myocardial infarction. ROS—reactive oxygen species, MMP2—metalloproteinase-2, MMP9—metalloproteinase-9, TIMP—tissue inhibitors of matrix metalloproteinase, ALDO—aldosterone, ANGII—angiotensin II, TGF-β—transforming growth factor beta, CTGF—connective tissue growth factor.

**Figure 2 jcm-11-05430-f002:**
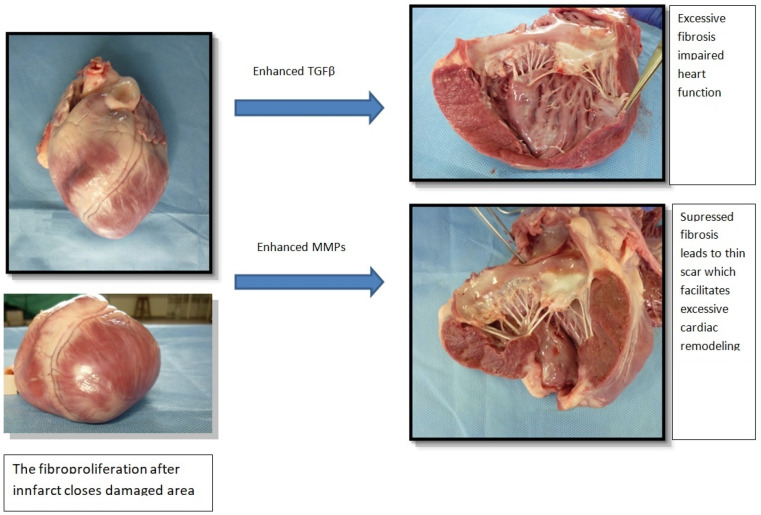
Fibroproliferation in the infarct zone. TGF-β—transforming growth factor beta, MMP—metalloproteinase.

**Figure 3 jcm-11-05430-f003:**
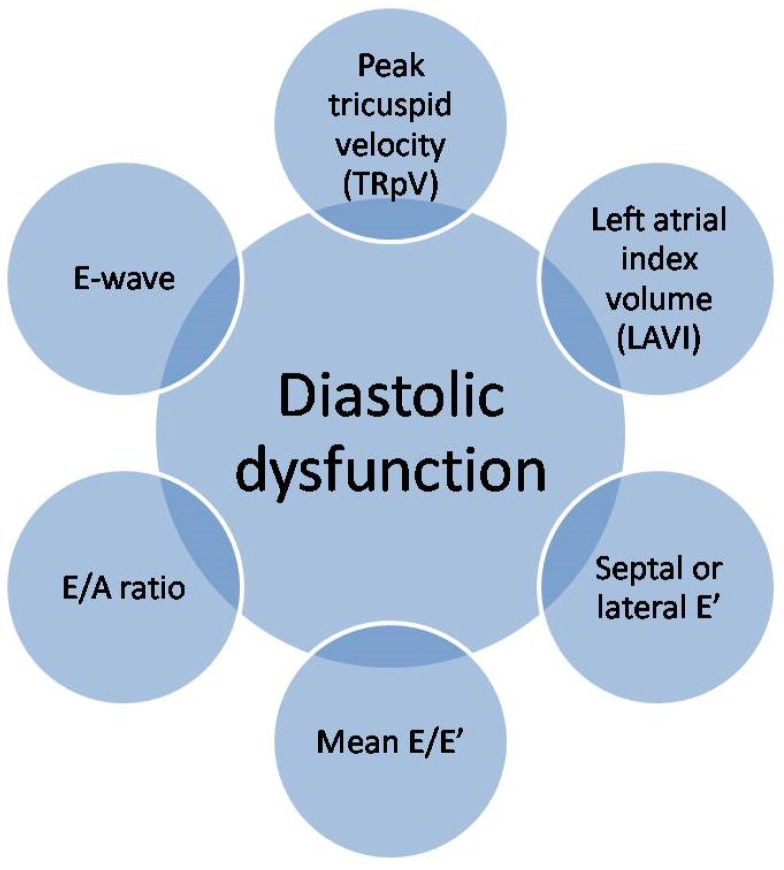
Echocardiographic assessment of left ventricular diastolic dysfunction.

**Figure 4 jcm-11-05430-f004:**
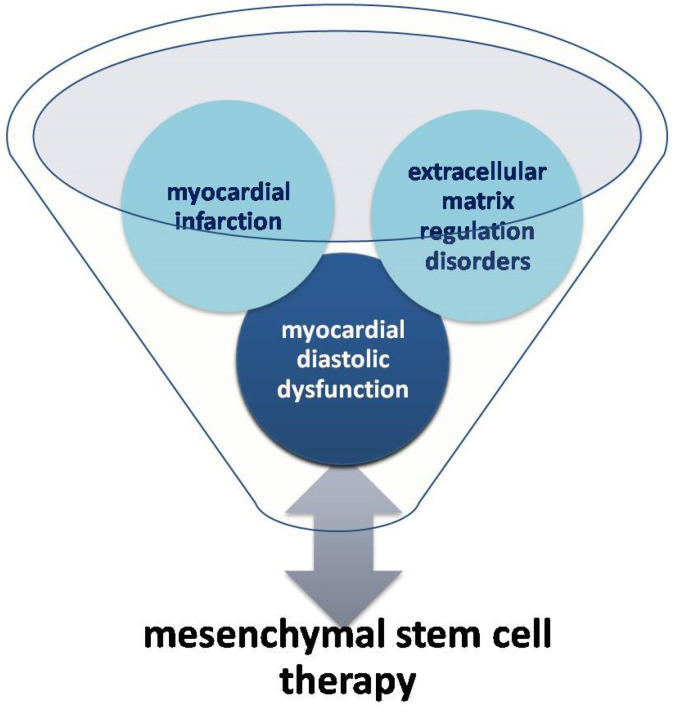
Relationship between mesenchymal stem cell therapy and myocardial diastolic dysfunction (Central Illustration).

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
