# Peer review of "Assessment of Myocardial Diastolic Dysfunction as a Result of Myocardial Infarction and Extracellular Matrix Regulation Disorders in the Context of Mesenchymal Stem Cell Therapy"

_jcm, 2022, doi:10.3390/jcm11185430_

Round 1

Reviewer 1 Report

In this short review, Patrycja Piątek-Matuszak et al focused on the mechanisms that ischemic cardiomyopathy brings to fibroproliferation and diastolic dysfunction. Secondly, they focused on stem cell therapy in this context.

Even if this is an interesting review, there are some comments to address:

Major comments

1)      Although the review begins with an introduction, it should also specify the objectives and how it intends to address one or more topics. So written, it seems disorganized, makes the reader lose, and fails to express the work behind it as well as the final message. In this regard, I recommend a reorganization and perhaps a division into parts of the review; I also suggest seeking the support of some other expert international colleagues on the subject in order to increase visibility and scientific value.

2)      In addition, the sentences range from basic English to complex and difficult-to-understand sentences. In this regard, an extensive review by native English is required.

3)      Moreover:

-          Page 2, lines 70-71: about types of collagens, 80% + 11% does not equal to 100%. Please, explain

-          Some sentences don’t have an end - page 6, lines 169-170: “Especially, if the diastolic dysfunction results from myocardial infarction and extracellular matrix regulation disorders.”

-          Page 7, lines 217-219 you need to put a reference for the statement.

Minor comments

1)      Ref 5 à typing error for Poland

2)      Page 1, line 42: abbreviations must be specified (ESC, ACCF, AHA, WHF)

3)      Page 4, line 105: the abbreviation “NIZ” should be specified before.

4)      Please check the spaces between the number of references in the text.

5)      Moderate English language revision is required by a native speaker

6)      Caption of figure 4 must be put at the bottom.

7)      Please specify what ATSC stands for

8)      Please, put a space between these two words: “Functionalrecovery” (page 6, line 188)

9)      Page 7, line 232: “[63].Using” a space is missing

10)   Page 8, line 256: “allows the design” à “allows TO design”

Author Response

Point-by-point response to reviewers

“Assessment of myocardial diastolic dysfunction as a result of myocardial infarction and extracellular matrix regulation disorders in the context of mesenchymal stem cell therapy”

__________________________________________________________________________

Response to Reviewer Comments

In this short review, Patrycja Piątek-Matuszak et al focused on the mechanisms that ischemic cardiomyopathy brings to fibroproliferation and diastolic dysfunction. Secondly, they focused on stem cell therapy in this context.

Even if this is an interesting review, there are some comments to address.

We are grateful for your comments.  I present the answers to the questions.

Regarding your comments, we have already made the necessary amendments to our manuscript.

Major comments

  • Although the review begins with an introduction, it should also specify the objectives and how it intends to address one or more topics. So written, it seems disorganized, makes the reader lose, and fails to express the work behind it as well as the final message. In this regard, I recommend a reorganization and perhaps a division into parts of the review; I also suggest seeking the support of some other expert international colleagues on the subject in order to increase visibility and scientific value.

Answer:             As suggested by the reviewer, the article has been divided into parts. We hope it will make it easier to read now.

  • In addition, the sentences range from basic English to complex and difficult-to-understand sentences. In this regard, an extensive review by native English is required.

Answer:             The text was checked by another native speaker – a cardiologist. The                                   previous, firstcorrection was made by a native speaker, but not a                                                      cardiologist.

3)      Moreover:

-          Page 2, lines 70-71: about types of collagens, 80% + 11% does not equal to 100%. Please, explain

Answer:     As we know, most of the collagen in the heart consists of collagen I (80%) and collagen III (11%), while the remaining forms of collagen represent only 9%. Information on other forms of collagen has been added to make the article easier for the reader to read.

-          Some sentences don’t have an end - page 6, lines 169-170: “Especially, if the diastolic dysfunction results from myocardial infarction and extracellular matrix regulation disorders.”

Answer:     This sentence was a continuation of the thoughts from the previous sentence. For easier understanding, the two sentences have been combined.

-          Page 7, lines 217-219 you need to put a reference for the statement.

Answer: The statement has been removed.

Minor comments

  • Ref 5 à typing error for Poland

Answer: The typing error has been corrected.

2)      Page 1, line 42: abbreviations must be specified (ESC, ACCF, AHA, WHF)

Answer: The abbreviations have been defined.

3)      Page 4, line 105: the abbreviation “NIZ” should be specified before.

Answer: It is defined in the article thus: “Increased fibrosis was observed both in the infarct zone (IZ) and in the non-infarct zone (NIZ).” It explains the acronym NIZ sufficiently extensively. Therefore, it is not described further.

4)      Please check the spaces between the number of references in the text.

Answer: Errors in spaces in the text have been corrected

5)      Moderate English language revision is required by a native speaker

Answer: The text has been corrected by a second native speaker – a cardiologist. The previous, first correction was made by a non-cardiologist.

6)      Caption of figure 4 must be put at the bottom.

Answer: The caption of Figure 4 has been placed below the picture.

7)      Please specify what ATSC stands for

Answer: The abbreviation has been defined.

8)      Please, put a space between these two words: “Functionalrecovery” (page 6, line 188)

Answer: The space has been added.

9)      Page 7, line 232: “[63].Using” a space is missing

 Answer: The space has been added.

10)   Page 8, line 256: “allows the design” à “allows TO design”

Answer: That would be incorrect. The sentence has been corrected by a native speaker.

Reviewer 2 Report

This paper by Piątek-Matuszak et al. reviewed changes in myocytes, extracellular matrix, and fibrosis after myocardial infarction (i.e. healing processes leading to ventricular remodeling). They provided an articulate account of phases of ventricular responses to acute myocardial infarction. They authors also reviewed stem cell therapies to regenerate myocytes and reduce fibrosis to recover extracellular matrix. Finally, they discussed literature on assessment of diastolic function and the impact of stem cell therapy. This is an informative review on current literature in the related areas. The following are my comments and suggestions.

1. Based on the title, this review would focus on diastolic function/dysfunction and assessment of diastolic function after myocardial infarction as a result of changes in extracellular matrix and the impact of stem-cell therapy. However, only a small portion of the paper is devoted to this (half of page 5). The authors may want to provide more information on: 1) what the mechanisms underlying diastolic dysfunction are after MI; 2) what clinical and physiological features of diastolic dysfunction are after MI, and 3) what the impact of stem cell therapy on improving diastolic function is (i.e. review and discuss published literature on improving diastolic function with stem cell therapy).  For example, the authors mentioned echocardiography as an important tool assessing diastolic function. But other means, such as cardiac MRI and cardiac catheterization, can be more accurate in terms of quantitative assessment and accurate diagnosis of diastolic dysfunction and assessment of cardiac fibrosis.

2. I would caution the authors in citing the work of Dr. Piero Anversa’s group (refs 33, 34) on stem cell therapy since much of these work were withdrawn due to fraud. When citing, the authors need to validate the findings from these publications.

3. A minor concern is that the literature cited on stem cell therapy are 5 years old. Any more recent papers published would bring the knowledge up to date in this area.

Author Response

Point-by-point response to reviewers

“Assessment of myocardial diastolic dysfunction as a result of myocardial infarction and extracellular matrix regulation disorders in the context of mesenchymal stem cell therapy”

__________________________________________________________________________

Response to Reviewer Comments

This paper by Piątek-Matuszak et al. reviewed changes in myocytes, extracellular matrix, and fibrosis after myocardial infarction (i.e. healing processes leading to ventricular remodeling). They provided an articulate account of phases of ventricular responses to acute myocardial infarction. They authors also reviewed stem cell therapies to regenerate myocytes and reduce fibrosis to recover extracellular matrix. Finally, they discussed literature on assessment of diastolic function and the impact of stem cell therapy. This is an informative review on current literature in the related areas. The following are my comments and suggestions.

We are grateful for your comments.  I present the answers to the questions.

 Regarding your comments, we have already made the necessary amendments to our manuscript.

  1. Based on the title, this review would focus on diastolic function/dysfunction and assessment of diastolic function after myocardial infarction as a result of changes in extracellular matrix and the impact of stem-cell therapy. However, only a small portion of the paper is devoted to this (half of page 5). The authors may want to provide more information on: 1) what the mechanisms underlying diastolic dysfunction are after MI; 2) what clinical and physiological features of diastolic dysfunction are after MI, and 3) what the impact of stem cell therapy on improving diastolic function is (i.e. review and discuss published literature on improving diastolic function with stem cell therapy). For example, the authors mentioned echocardiography as an important tool assessing diastolic function. But other means, such as cardiac MRI and cardiac catheterization, can be more accurate in terms of quantitative assessment and accurate diagnosis of diastolic dysfunction and assessment of cardiac fibrosis.

Answer: The paper has been modified in accordance with the comments of the reviewer.

  1. I would caution the authors in citing the work of Dr. Piero Anversa’s group (refs 33, 34) on stem cell therapy since much of these work were withdrawn due to fraud. When citing, the authors need to validate the findings from these publications.

Answer: Thank you for the warning. After discussing it with the team of authors, we decided to delete this sentence. We agreed that this could be done without losing the sense of the information we wanted to convey.

  1. A minor concern is that the literature cited on stem cell therapy are 5 years old. Any more recent papers published would bring the knowledge up to date in this area.

Answer:The literature has been updated. We are grateful to the reviewer for the effort of evaluating the paper and contributing to its improvement.

Round 2

Reviewer 1 Report

the authors addressed all the comments